# Staying Alive: Uncensored Survival Analysis with Tabular Foundation Models

**Mariana Vargas Vieyra** [1]

## Abstract

Survival Analysis (SA) is a statistical framework that models the time span until some event of interest occurs. Widely used in several domains, including healthcare and churn prediction, a central challenge in its applicability stems from the time of the event being partially observed or *right-censoring*. Tabular Foundation Models (TFM) have attracted significant interest in recent years due to their ability to perform prediction tasks in a single forward pass, requiring no dataset-specific parameter fitting. Despite their success, their application to prediction tasks on time-to-event data remains difficult due to right censoring. In this work, we present a training-free method to survival regression by leveraging TFMs to both predict the time of the event and iteratively impute right-censored data. Our method uses a TFM to construct an Accelerated Failure Time (AFT) model requiring no training beyond fitting a single scalar parameter. Subsequently, by building on the Buckley-James estimator, we introduce a non-parametric in-context estimator for right-censored data. Our experiments on standard survival analysis benchmarks show that our method is competitive with several parametric and semi-parametric survival regression models that require training, including Cox regression and parametric AFT models[1].

## 1. Introduction

Survival Analysis (SA) (Kalbfleisch & Prentice, 2002) is a family of statistical procedures that aim to model the time from entry into a study and the occurrence of an event of interest, commonly referred to as *time-to-event* data. Applications of SA span numerous domains, such as healthcare (Lee & Go, 1997), churn prediction (Kvamme et al., 2019; Ren et al., 2019), and time to failure of mechanical systems (Papathanasiou et al., 2023). In practice, time-to-event data is most commonly presented in tabular form, requiring backbone models that can handle tabular inputs natively.

Tabular Foundation Models (TFM) (Hollmann et al., 2023; 2025; Qu et al., 2025; 2026) have emerged as a promising paradigm for tabular data prediction tasks due to their ability to perform in-context classification and regression in a single forward pass with no dataset-specific training. However, they have limited applicability to survival regression due to *right-censoring*. Right-censoring is a ubiquitous phenomenon in SA: some subjects leave the study or reach the study horizon before the event occurs, leaving their event time unobserved. While classical survival models naturally account for censored data, TFMs expect fully observed prediction targets, rendering survival regression difficult. This problem requires sophisticated imputation strategies. Parner & Andersen (2010) calculate pseudo-targets via jackknife leave-one-out iterations of the Kaplan-Meier estimator, transforming right-censored data into an unbiased and complete dataset. A principled solution was introduced by Buckley & James (1979), who proposed to calculate pseudo-targets for censored data by iteratively estimating the conditional expectation of the censored event time under the current model and updating the model accordingly. More recently, Kim et al. (2026) introduced a TFM based algorithm that discretizes the time range and frames survival regression as a classification problem where the task is to predict the survival outcome in each bin. Even though this model can accommodate censored data, its performance is sensitive to the level of discretization, with coarser binning degrading the performance of the model.

In this work we propose to frame survival regression as a prediction task and leverage TFMs for zero-shot survival prediction, introducing two complementary contributions. We first introduce a mechanism for constructing an Accelerated Failure Time model with a single scalar parameter by leveraging a TFM. To address censoring, we propose to leverage a TFM as a non-parametric in-context estima-

---

[1]Rhizome Labs, Paris, France. Correspondence to: Mariana Vargas Vieyra <mariana.vargas@rhizome-labs.com>.

*Proceedings of the 2nd ICML Workshop on Foundation Models for Structured Data*, Seoul, South Korea. 2026. Copyright 2026 by the author(s).

[1]Code at https://github.com/marianaw/frozenhazard.git.

tor for imputing censored data. Through experiments on five widely used survival benchmarks we demonstrate that our method achieves comparable performance to classical parametric and semi-parametric survival models that require training.

## 1.1. Background

**Survival analysis and the AFT model.** We consider datasets of the form $\{(t_i, \Delta_i, \mathbf{x}_i)\}_{i=1}^N$, where $t_i = \min(T_i, C_i)$ is the observed time, $T_i$ is the event time, $C_i$ is the censoring time and $\Delta_i = \mathbf{1}(\tilde{t}_i \leq C_i)$ indicates whether the event was observed for subject $i$.

In the Accelerated Failure Time (AFT) model, the logarithm of the event time is modeled linearly as

$$\log T_i = \mathbf{x}_i^\top \beta + \sigma \epsilon_i, \tag{1}$$

where $\epsilon_i \overset{\text{i.i.d.}}{\sim} \mathcal{N}(0, 1)$ and $\mathbf{x}_i$ is the vector of features characterizing subject $i$. Denoting $\mu_i = \mathbf{x}_i^\top \beta$, the corresponding survival function is

$$\hat{S}(t \mid \mathbf{x}_i) = 1 - \Phi\left(\frac{\log t - \mu_i}{\sigma}\right), \tag{2}$$

where $\Phi$ denotes the standard Gaussian cumulative distribution function.

Under right censoring, the log-likelihood is given by

$$
\begin{aligned}
\ell(\sigma) = \sum_{\Delta_i = 1} &\left[\log \phi\left(\frac{\log t_i - \mu_i}{\sigma}\right) - \log \sigma - \log t_i\right] \\
&+ \sum_{\Delta_i = 0} \log\left[1 - \Phi\left(\frac{\log t_i - \mu_i}{\sigma}\right)\right].
\end{aligned}
\tag{3}
$$

where $\phi$ and $\Phi$ denote the standard Gaussian density and cumulative distribution functions respectively.

**Buckley-James estimator.** The Buckley-James estimator is an iterative regression technique for censored survival data that replaces each censored observation with its conditional expected value. That is, it computes $Y_i^* = \mathbb{E}[\log T_i \mid \log T_i > \log C_i, \mathbf{x}_i, \beta]$ for censored subjects. This method combines the current model's prediction with the Kaplan-Meier estimates of the residual distribution, allowing for a standard least-squares fit on the "completed" dataset. More specifically, at each iteration, it defines the targets as:

$$
Y_i^* = \begin{cases}
\log t_i & \text{if } \Delta_i = 1 \\
\hat{\mu}_i + \dfrac{\displaystyle\sum_{j: \tilde{e}_j > \tilde{e}_i, \Delta_j = 1} \tilde{e}_j \cdot w_j}{1 - \hat{F}_\epsilon(\tilde{e}_i)} & \text{if } \Delta_i = 0
\end{cases}
$$

where $\tilde{e}_i = \log t_i - \hat{\mu}_i$ is the observed residual, $\hat{F}_\epsilon$ is the Kaplan-Meier estimate of the residual distribution fitted on $\{(\tilde{e}_i, \Delta_i)\}_{i=1}^N$, and $w_j$ are the probability masses assigned by $\hat{F}_\epsilon$ at each uncensored residual. At its core, the Buckley-James procedure iteratively solves the following fixed point equation: $\hat{\beta} = (\mathbf{X}^\top \mathbf{X})^{-1} \mathbf{X}^\top \mathbf{Y}^*(\hat{\beta})$.

## 2. Survival Regression via In-Context Learning

The goal of this work is to present a method for leveraging TFMs to perform survival regression without dataset-specific training. To this end, we frame survival regression as an in-context prediction task.

Let $(X_{\text{tr}}, \mathbf{t}_{\text{tr}}, \mathbf{\Delta}_{\text{tr}})$ denote the training set and $X_{\text{test}}$ the test set. We further define $\mathcal{U} = \{i : \Delta_i = 1\}$ and $\mathcal{C} = \{i : \Delta_i = 0\}$ as the set of uncensored and censored training instances respectively, and let $f$ denote a TFM used as a regressor backbone. We then estimate the AFT model defined in Equation (1) by regressing the target through in-context learning:

$$\hat{\mu} = f(\{\mathbf{x}_i, \log t_i\}_{i \in \mathcal{U}}, \ X_{\text{test}}).$$

Once $\hat{\mu}$ is obtained, we estimate $\sigma$ by maximizing the log-likelihood in Equation (3):

$$\hat{\sigma} = \arg\max_\sigma \ \ell(\sigma; \hat{\mu}, \mathbf{t}_{\text{tr}}, \mathbf{\Delta}_{\text{tr}}),$$

over the full training set. Observe that $\sigma$ is a scalar value and the sole trainable parameter in our model.

This algorithm provides an effective way of leveraging TFMs for survival regression. However, restricting the training context to uncensored observations introduces bias, as samples with longer survival times are disproportionately censored, leading the model to underestimate survival times. To overcome this problem we propose to calculate pseudo-targets for censored data via an iterative mechanism analogous to the Buckley-James estimator.

### 2.1. Imputing censored data with TFMs

The core idea is to use a TFM as a non-parametric in-context estimator, iteratively imputing survival times as pseudo-targets. Pseudo-targets are initialized with a data-driven warm start based on the Kaplan-Meier jackknife estimator. We then iteratively refine both the pseudo-targets and scale parameter $\sigma$.

**Initialization.** Let $t_0$ be the median observed event time and $\tilde{\theta}_i$ the jackknife pseudo-observation of subject $i$ at $t_0$, obtained via leave-one-out Kaplan-Meier estimation. We obtain a warm-start scale estimate $\hat{\sigma}^{(0)}$ by maximizing Equation (3) using pseudo-observation-based predictions. Cen-

sored pseudo-targets are then initialized as:

$$Y_i^{*(0)} = \max\Big(\log t_i, \ \log t_0 - \hat{\sigma}^{(0)} \cdot \Phi^{-1}(1 - \tilde{\theta}_i)\Big) \ ,$$

for $i \in \mathcal{C}$. This formulation inverts the AFT survival function at $t_0$ to map pseudo-targets on the survival probability scale back to imputed log-times. The $\max$ operation enforces $T_i > C_i$.

**Pseudo-targets and scale updates at iteration $k$.** At each iteration $k$, we form a stochastic context $\mathcal{U}^{(k)} \cup \mathcal{C}^{(k)}$, with $\mathcal{U}^{(k)} \subseteq \mathcal{U}$ and $\mathcal{C}^{(k)} \subseteq \mathcal{C}$, by sub-sampling censored and uncensored subjects from the training set. This ensures that each censored subject is predicted out-of-sample, preventing degenerate self-prediction. We then make a forward pass with the TFM to predict the mean log-time of the censored subjects that were not included in the context:

$$\hat{\boldsymbol{\mu}}_{\mathcal{C}\setminus\mathcal{C}^{(k)}}^{(k)} = f\bigg(\Big\{\mathbf{x}_j, \ Y_j^{*(k-1)}\Big\}_{j \in \mathcal{U}^{(k)} \cup \mathcal{C}^{(k)}}, \ X_{\mathcal{C}\setminus\mathcal{C}^{(k)}}\bigg)$$

Once the mean log-times are predicted, we update the pseudo-targets of the training set as follows:

$$Y_i^{*(k)} = \begin{cases} \log t_i & i \in \mathcal{U} \\ \max\Big(\log t_i, \ \hat{\mu}_i^{(k)} + \hat{\sigma}^{(k-1)} \cdot \epsilon_i^{(k)}\Big) & i \in \mathcal{C} \end{cases} \ ,$$

where $\epsilon_i^{(k)} \overset{\text{i.i.d.}}{\sim} \mathcal{N}(0,1)$. Unlike the Buckley-James estimator, which imputes the conditional mean of censored survival times, our method updates are drawn from a truncated distribution $p(\log T_i | T_i > C_i; \hat{\mu}_i^{(k)}, \hat{\sigma}^{(k-1)}$, enforcing the constraint that $T_i > C_i$ for censored subjects. Finally, we update the scale parameter by MLE on the full training set, both censored and uncensored subjects alike.

$$\hat{\sigma}^{(k)} = \arg\max_{\sigma} \ell\Big(\sigma; \ \hat{\boldsymbol{\mu}}_{\text{tr}}^{(k)}, \ \mathbf{t}_{\text{tr}}, \ \boldsymbol{\Delta}_{\text{tr}}\Big) \qquad (4)$$

We repeat this procedure until the pseudo-targets stabilize, that is, $\|Y^{*(k)} - Y^{*(k-1)}\|_2 < \varepsilon$, or a maximum number of iterations is reached.

**Inference.** At test time, we make a forward pass to obtain the mean log-times of the test set:

$$\hat{\boldsymbol{\mu}}_{\text{test}} = f\bigg(\Big\{\mathbf{x}_j, \ Y_j^{*(K)}\Big\}_{j \in \mathcal{U} \cup \mathcal{C}}, \ X_{\text{test}}\bigg) \ ,$$

where $K$ is the last iteration. The survival function for the test set is given by (2) on $\hat{\boldsymbol{\mu}}_{\text{test}}$ and $\hat{\sigma} = \hat{\sigma}^{(K)}$, the final fitted scale parameter.

## 3. Experiments

In this section, we empirically evaluate the ability of TFMs to perform survival analysis without dataset-specific training. We compare TFM-based approaches against classical

survival models across five benchmark datasets. Among the former, we evaluate three variants of our method, TABSA, spanning settings from naive complete case scenarios to our iterative imputation method, and the method proposed by Kim et al. (2026) (TabSA-Bin). Finally, we present an ablation study on the number of bins used by BinSFA, analyzing the effect of discretization granularity on predictive performance.

**Datasets.** We consider five publicly available survival analysis benchmarks spanning cardiovascular disease, oncology and critical care. These include **WHAS500** (Hosmer et al., 2008), which studies post-heart-attack survival in 500 patients, **GBSG** (Schumacher et al., 1994) and **METABRIC** (Curtis et al., 2012), two breast cancer datasets comprising 686 and 1,903 patients respectively, **SUPPORT** (Knaus et al., 1995), containing approximately 8873 critically ill patients, and **FLCHAIN** (Dispenzieri et al., 2012), a serum biomarker study with approximately 7874 subjects. For all datasets we standarize continuous variables and report results over 10 random seeds with 80/20 train-test splits. In Table 2 in Appendix C we present summary statistics of the datasets.

**Baselines.** We compare against four classical survival models fitted from scratch on each split: Cox Proportional Hazards (Cox PH) (Cox, 1972), Weibull and Log-Normal AFT (Wei, 1992), and Random Survival Forest (RSF) (Ishwaran et al., 2008). We also include **TabSA-Bin** (Kim et al., 2026) as the primary zero-shot competitor. TabSA-Bin works by re-framing survival analysis as a binary classification problem where the time range is split into $K$ event-time quantiles. A TFM then predicts, for each bin, whether the event of interest occurred or not, and assembles a step-function survival curve. Despite being an effective mechanism for addressing right-censoring in survival data, we argue that the discretization of time introduces a resolution-performance tradeoff. Finer grids better approximate the continuous survival function but reduce the effective context size at late quantiles, while coarser grids preserve context at the cost of approximation accuracy. Our method operates in continuous time and is therefore free of this tradeoff.

**Proposed Method.** We evaluate three instantiations of our framework, representing a progression from simple scenarios to a principled imputation mechanism. We first assume a Complete Case Analysis (CCA) setup. That is, we regress the log-time of uncensored subjects only, dismissing censored observations as missing data (TabSA-CCA). We also consider an intermediate method that addresses censoring via KM jackknife imputation prior to regression, serving as an ablation of the full iterative procedure (TabSA-PO). Finally, we benchmark our iterative imputation mechanism

*Table 1.* C-index ($\uparrow$) and IBS ($\downarrow$) across five benchmarks (mean $\pm$ std, 10 splits). **Bold**: best per column overall; underline: best zero-shot method per column. ‡TabSA-PO (TabICL) degenerates on these datasets (C-index $\approx 0.5$).

| Method | WHAS500 | | GBSG | | METABRIC | | SUPPORT | | FLCHAIN | |
|---|---|---|---|---|---|---|---|---|---|---|
| | C-idx | IBS | C-idx | IBS | C-idx | IBS | C-idx | IBS | C-idx | IBS |
| *Classical (trained)* | | | | | | | | | | |
| Cox PH | **0.768**±.036 | **0.168**±.025 | 0.661±.029 | 0.190±.009 | **0.643**±.020 | **0.185**±.010 | 0.564±.007 | 0.212±.003 | 0.724±.009 | 0.100±.003 |
| Weibull AFT | **0.769**±.031 | **0.169**±.025 | 0.662±.031 | 0.191±.009 | **0.643**±.020 | **0.185**±.011 | 0.563±.007 | 0.213±.004 | **0.797**±.009 | 0.099±.003 |
| Log-Normal AFT | 0.767±.030 | 0.171±.021 | 0.669±.033 | **0.188**±.009 | 0.646±.017 | 0.187±.009 | 0.570±.007 | 0.214±.004 | **0.797**±.008 | 0.101±.004 |
| RSF | 0.745±.031 | **0.162**±.013 | **0.677**±.033 | **0.185**±.012 | 0.624±.029 | 0.190±.013 | **0.616**±.006 | **0.197**±.003 | 0.725±.008 | 0.100±.003 |
| *Zero-shot, TabPFN backbone* | | | | | | | | | | |
| TabSA-CCA (ours) | 0.699±.046 | 0.311±.041 | 0.656±.031 | 0.272±.029 | 0.619±.020 | 0.229±.011 | 0.553±.008 | 0.245±.006 | 0.665±.021 | 0.234±.002 |
| TabSA-Bin | **0.779**±.038 | **0.166**±.019 | 0.673±.017 | **0.190**±.014 | **0.642**±.017 | **0.184**±.011 | 0.611±.008 | 0.201±.004 | 0.715±.011 | **0.095**±.003 |
| TabSA-PO (ours) | 0.740±.030 | 0.182±.019 | 0.654±.024 | 0.202±.016 | 0.637±.024 | 0.206±.016 | **0.614**±.008 | 0.201±.005 | 0.714±.008 | 0.106±.004 |
| TabSA-BJ (ours) | 0.776±.037 | 0.191±.027 | **0.680**±.020 | 0.220±.020 | **0.651**±.019 | 0.202±.010 | 0.610±.009 | 0.216±.006 | 0.791±.009 | 0.136±.003 |
| *Zero-shot, TabICL backbone* | | | | | | | | | | |
| TabSA-CCA (ours) | 0.750±.026 | 0.267±.039 | 0.662±.029 | 0.256±.028 | 0.624±.017 | 0.216±.010 | 0.573±.008 | 0.220±.004 | 0.661±.018 | 0.238±.002 |
| TabSA-Bin | **0.779**±.033 | 0.165±.018 | **0.675**±.014 | **0.189**±.014 | **0.642**±.017 | 0.185±.010 | 0.610±.007 | 0.201±.004 | 0.717±.014 | **0.095**±.004 |
| TabSA-PO (ours) | 0.767±.031 | 0.202±.017 | 0.504±.011‡ | 0.227±.014 | 0.547±.013‡ | 0.248±.010 | 0.608±.007 | 0.215±.003 | 0.568±.008‡ | 0.115±.004 |
| TabSA-BJ (ours) | 0.780±.038 | 0.191±.025 | **0.687**±.027 | 0.217±.022 | 0.645±.018 | 0.207±.011 | 0.608±.007 | 0.223±.006 | 0.777±.011 | 0.121±.003 |

inspired by the Buckley-James estimator, TabSA-BJ, that alternates between refining censored pseudo-times and re-estimating $\sigma$ over 10 iterations. For our experiments we fix the proportion of sub-sampled censored data to 50%. We use two publicly available TFM backbones, TabPFN (Hollmann et al., 2023) and TabICL (Qu et al., 2025).

We report Harrell's C-index ($\uparrow$) and the IPCW Integrated Brier Score (IBS, $\downarrow$), integrated from the 5th to 95th percentile of observed training times (Graf et al., 1999). Metrics are describedin Appendix B. Results are presented in Table 1 as mean $\pm$ standard deviation across ten random splits. Results that are significantly superior under a paired $t$-test at significance level $\alpha = 0.05$ are highlighted in bold, while the best-performing TFM-based approach is underlined.

### 3.1. Results

Results show that TabSA-BJ improves over its simpler variants on C-Index, thus validating the iterative imputation approach. Notably, it achieves the largest improvement margins on the datasets with higher censoring rates. In terms of IBS, all continuous-time TabSA variants lag behind TabSA-Bin, suggesting that discretization may be better suited for calibration. Our method, by contrast, achieves stronger discrimination as measured by C-index. We argue that TabSA-Bin and TabSA-BJ are complementary mechanisms: the former optimizes survival probabilities at fixed quantiles, making it naturally suited for calibration (low IBS), whereas the latter operates in log-time scale, thus preserving the rank structure that drives a high C-Index. While the best IBS scores are often achieved by classical methods, we observe that our method often outperforms or is on a par with the best scoring algorithms in terms of C-Index, despite requiring no dataset-specific training.

Both backbones achieve comparable performance overall, although TabICL yields degenerate predictions for TabSA-PO on three datasets. We leave a deeper investigation of this behavior for future work.

**Convergence analysis.** Figure 1 in Appendix A illustrates the convergence behavior of TabSA-BJ across iterations. We observe that $\sigma$ converges to a stable value, indicating that the pseudo-targets have reached a fixed point. Notably, on several datasets $\sigma$ converges to values close to 1, suggesting that the TFM backbone captures a substantial fraction of the variance in log-time under the AFT model. This hints at the capacity of tabular foundation models to encode structure relevant to survival regression without explicit survival-specific training. As $\sigma$ stabilizes, the C-Index values attain their best score, while the IBS tends to reach a minimum and then increase slightly. This suggests that early stopping may be beneficial for calibration even if discrimination continues to improve. We leave the tuning of the number of iterations as future work.

## 4. Conclusion

We introduced TabSA, a framework for zero-shot survival regression that leverages frozen tabular foundation models as regression backbones, requiring no dataset-specific training beyond fitting a single scalar parameter. To handle right-censoring, we proposed TabSA-BJ, an iterative procedure inspired by the Buckley-James estimator, which progressively refines pseudo-targets for censored subjects via in-context imputation. Experiments across five benchmarks demonstrate that TabSA-BJ is competitive with classical trained survival models on discrimination. Our results suggest that TFMs encode sufficient prior knowledge about regression structure to generalize to survival tasks with minimal adaptation, opening a promising direction for zero-shot statistical modeling on censored data.

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

## A. Convergence of TabSA-BJ

Figure 1 illustrates how the scalar parameter $\sigma$ stabilizes, indicating our method successfully converges to a fixed point. Note how the IBS reaches a minimum in early iterations, suggesting our method would benefit from tuning of the number of iterations.

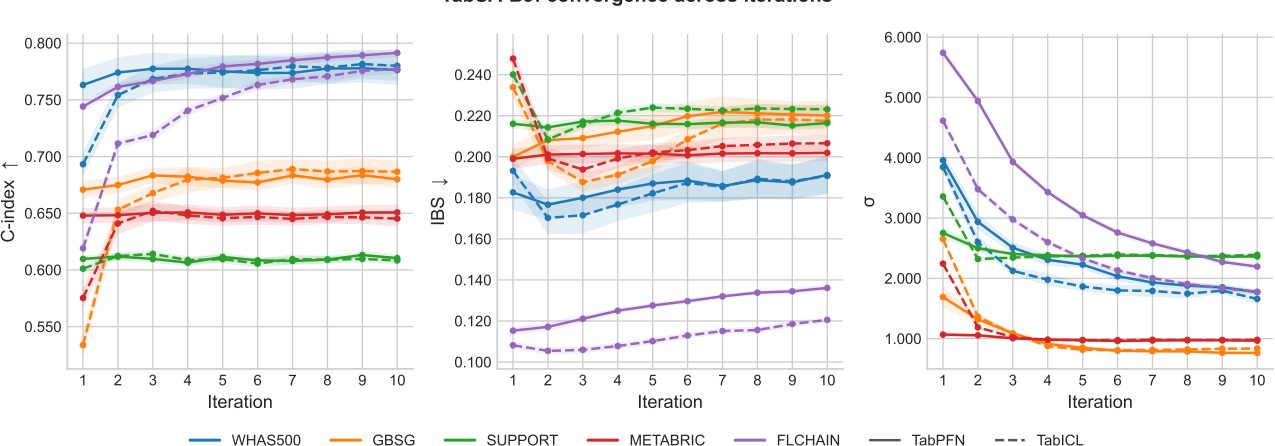

*Figure 1.* Convergence of IBS, C-Index and $\sigma$ scalar parameter for all datasets and TFM backbones.

## B. Evaluation Metrics

We evaluate all models using two standard survival metrics, Harrel's C-Index, and IPCW Integrated Brier Score. We provide details for both of these below.

The Harrel's C-Index, also known as *Concordance Index* or just C-Index, is a goodness of fit measure for risk models. Defined as

$$\mathrm{CI}(\mathcal{D}) = P\big\{S(t_i|\mathbf{x}_i) > S(t_j|\mathbf{x}_j) : t_i > t_j \text{ and } \Delta_j = 1\big\}.$$

where $\mathcal{D} = (X, \mathbf{t}, \boldsymbol{\Delta})$ is the dataset, it measures how well the model ranks subjects according to their predicted risk scores. A C-Index close to $1$ indicates the model ranks subjects almost perfectly, whereas a value near $0.5$ is a symptom the model is ranking at random.

Another metric is based on the Brier Score, which functions as a squared error of the estimated survival times with respect to the true survival outcome of the subjects. Integrating these scores over time gives the Integrated Brier Score. However, in datasets with strong censoring, many survival outcomes remain unknown. To account for censored data, one can calculate the Inverse Probability of Censoring Weighting Brier Score. This metric ponders subjects by the inverse probability of them not being censored. It is defined as:

$$\mathrm{BS}(t) = \frac{1}{N}\sum_{i=1}^{N}\left[\frac{(0 - S(t \mid \mathbf{x}_i))^2 \cdot \mathbb{I}(t_i \leq t, \Delta_i = 1)}{G(t_i)} + \frac{(1 - S(t \mid \mathbf{x}_i))^2 \cdot \mathbb{I}(t_i > t)}{G(t)}\right]$$

where $G(t) = p(C > t)$ is the Kaplan-Meier estimator of the censoring distribution.

## C. Summary Statistics of Datasets

The following table summarizes the main statistics of the datasets used in the experimental section.

*Table 2.* Summary statistics of the datasets used in the experiments.

| DATASET | SAMPLES | FEATURES | CENSORING |
|---------|---------|----------|-----------|
| WHAS500 | 500 | 14 | 57.0% |
| GBSG | 686 | 8 | 56.4% |
| METABRIC | 1,904 | 9 | 42.0% |
| SUPPORT | 8,873 | 14 | 31.9% |
| FLCHAIN | 7,874 | 9 | 72.5% |

