# OpenReview forum: "Staying Alive: Uncensored Survival Analysis with Tabular Foundation Models"
_ICML.cc/2026/Workshop/FMSD — FMSD @ ICML 2026 Poster_

### Official Review · Reviewer_tiSB · 2026-05-13

**Rating:** 6
**Confidence:** 4

**Review:**

**Summary** The work proposes using Tabular Foundation Models (TFMs) to construct an Accelerated Failure Time (AFT) model for survival regression without additional training or fine-tuning.

**Strengths**:

- The paper contains comprehensive baselines, including four classical survival models fitted from scratch on each split.

- The models are evaluated on five publicly available survival analysis benchmarks spanning diverse healthcare domains of cardiovascular disease, oncology and critical care

- The work evaluates models in three scenarios: uncensored subjects only, with censored subjects using KM jackknife imputation, and with censored subjects using iterative imputation mechanism inspired by the Buckley-James estimator.

- The paper is well-organized and easy to follow with methods outlined mathematically.

**Areas for Improvement:**

- Most of the classical baselines handle censoring natively and do not require target imputation. It is unclear whether these baselines are evaluated on the original censored data or the imputed targets, which should be explicitly stated. An ablation studying the impact of target imputation on such models would help demonstrate how such imputation impacts model calibration. Significant degradations would raise concerns about the validity of imputation.

- The proposed model seems to notably underperform baselines for specific datasets in terms of IBS (ex., WHAS500) which warrants discussion.

- Results would benefit from stating quantitative results such as percentage improvements.

- The methods are outlined clearly mathematically. The work could benefit from outlining the methods (TFM application, AFT model, and imputation strategies) with figures as well.

- Discussion of the motivation for the AFT model, its assumptions, and how they differ from the proportional hazards assumption underlying Cox PH would be beneficial for readers.

**Justification of Score** The paper presents a novel application of TFMs for survival regression, differentiating itself from prior work through its modeling strategy (AFT model) and mechanism for handling right-censored data (imputation). The paper is well-organized and would be relevant to readers interested in tabular foundation models and survival regression. However, I remain skeptical of target imputation as a mechanism for handling right-censoring, and believe such techniques warrant a more controlled study to assess their impact on model calibration which is not studied or discussed in this paper. For example, the proposed model seems to notably underperform baselines for specific datasets in terms of IBS (ex., WHAS500) which warrants discussion and the results section does not provide quantitative assessments.

---

### Official Review · Reviewer_o2rE · 2026-05-19
**Interesting but based on which "prior"?**

**Rating:** 6
**Confidence:** 4

**Review:**

Summary:
Frame survival regression as a prediction task and leverage Tabular Foundation Models (TFMs) for zero-shot survival prediction

Strengths:
- The intent is interesting and is the subject of recent research (see https://arxiv.org/abs/2603.29475, https://arxiv.org/pdf/2605.15488): trying to do zero-shot survival predict
- Creative, good engineering
- Code is provided

Areas for improvement:
- Timings? Running on CPU or GPU? Versus other models (non-TFMs), are these models worth the hype?
- TFMs are not pretrained on survival data, how can we explain the transfer? You just say: surprisingly, it works
- Have you considered pretraining your model directly on survival data?

---

### Official Review · Reviewer_CrHt · 2026-05-20
**Ideas to use TFMs for Survival Analysis, but no significant experimental results**

**Rating:** 5
**Confidence:** 4

**Review:**

**Summary**

The paper focuses on Survival Analysis, which models the probabilistic distribution of time-to-event data. Following Accelerated Failure Time models, the authors model the problem as a regression with gaussian noise of the log-time over provided features. One challenge of this task is the presence of right-censored data: subjects which have left before the end of the experiment.

Typically, a Buckley-James estimator is used to estimate the expected conditional time of censored series, which allows to iteratively estimate the parameters of the regression. Inspired by this method, the authors propose to use Tabular Fondation Models for estimating subjects' expected time. The authors evaluate different procedures: regressing only on non-censored data (TabSA-CCA), imputing the censored data before regressing (TabSA-PO), the alternated estimation of expected times and noise amplitudes (TabSA-BJ). In their method, the authors also propose a sampling procedure to include sensored subjects in the training sets, and use a truncated distribution to enforce a constraint of estimated time greater than censored time.

Experiments are run on multiple datasets with varying degree of censoring, multiple seeds and two state-of-the-art tabular foundation models (TabPFN and TabICL).

**Strengths**

The authors propose sensible improvements to the original BJ procedure, notably to extend beyond linear modeling using the pretrained foundation model. The plots illustrating the convergence of the estimated sigma show their proposed iterated method does converge, and the obtained results show pretrained models can perform as well as typical trained baselines. Experimenting on multiple backbone models and using multiple datasets with varying degree of censorship enables interesting additional analysis.

**Areas of improvements**

Results do not seem convincing enough to claim that their method is "competitive" against baselines. A few typos as well.

**Detailed comments**

The main results in Table 1 do not seem sufficient to conclude the propose method perform well on this task. The TabICL backbone fails on the last three datasets and in most settings either TabSA-Bin (method from another paper) or classical baselines perform better, especially on the IBS metric which is relevant in survival analysis. Furthermore, the claim that TabSA doesn’t require training is not exactly fair, since there is an iterative fitting of the noise.

Comments on form / typos:

- In equation (1), x_i is not defined. I assume they correspond to abstract features, but they are never described for each dataset
- For clarity perhaps the survival function should be introduced as S(t)=P(T>t) first, before giving its exact expression
- Similarly Y_i* should be defined (e.g $Y_i^* = \mathbb{E}[\log T_i \mid \log T_i > \log t_i, x_i]$)
- Line 122 second column: “10 random “ (splits?)
- Line 187: “Figre1”
- I believe the log-likelihood in equation 3 might be missing certain terms. Shouldn’t it be:

 $\ell(\sigma, \beta) = \sum_{\Delta_i=1} \left[ \log \phi\left(\frac{\log t_i - \mu_i}{\sigma}\right) - \log \sigma - \log t_i \right] + \sum_{\Delta_i=0} \log \left[ 1 - \Phi\left(\frac{\log t_i - \mu_i}{\sigma}\right) \right]$

- Maybe we could order the columns of Table 1 by degree of censoring, for easier analysis
- I would remind the definitions of the two metrics in appendix, and their differences for better interpretation

**Justification of score**

The framing of the problem via tabular regression using foundation model is interesting but the results do not allow to conclude on the relevance of this approach.